# Synergistic Effects of 2-Butyne-1,4-Diol and Chloride Ions on the Microstructure and Residual Stress of Electrodeposited Nickel

**DOI:** 10.3390/ma16093598

**Published:** 2023-05-08

**Authors:** Ming Sun, Chao Zhang, Ruhan Ya, Hongyu He, Zhipeng Li, Wenhuai Tian

**Affiliations:** School of Materials Science and Engineering, University of Science and Technology Beijing, Beijing 100083, China; dxsm0526@163.com (M.S.); zhang_chaovip@163.com (C.Z.);

**Keywords:** electrodeposition, additive, chloride ion, sulfamate, residual stress

## Abstract

To assess the individual and synergistic effects of 2-butyne-1,4-diol (BD) and chloride ions on the microstructure and residual stress of electrodeposited nickel, various nickel layers were prepared from sulfamate baths comprising varying concentrations of BD and chloride ions by applying direct-current electrodeposition. And their surface morphologies, microstructure, and residual stress were tested using SEM, XRD, EBSD, TEM, and AFM. While the nickel layers composed of pyramid morphology were prepared from additive-free baths, the surface flattened gradually as the BD concentration of the baths was increased, and the acicular grains in the deposits were replaced with <100> oriented columnar grains or <111> oriented nanograins; additionally, the residual tensile stress of the deposits increased. The addition of chloride ions to the baths containing BD significantly increased the residual stress in the nickel layers, although it only slightly promoted surface flattening and columnar grain coarsening. The effects of BD and chloride ions on the growth mode and residual stress of nickel deposits were explained via analysis of surface morphologies and microstructure. And the results indicate that the reduction of chloride ion concentration is a feasible way to reduce the residual stress of the nickel deposits when BD is included in the baths.

## 1. Introduction

Electrodeposition technology has been widely employed in many fields, such as coatings, electronics, and weapons, due to its ease of controlling the microstructure and properties of deposited materials by altering the electrolyte composition and electrodeposition parameters [1,2,3,4]. Nickel, which has excellent corrosion resistance and mechanical properties, is one of the most extensively used electrodeposited metals [5,6]. Electrodeposited nickel, particularly using sulfamate baths, has good mechanical properties, low residual stress, and high deposition rates and is very suitable for preparing thick nickel deposits required by the microfabrication and electroforming industry [7,8,9]. For thick nickel deposits, low residual stress and high toughness are required to prevent its cracking or bulging as well as deformation during and after the electrodeposition [10,11]. Additionally, the high residual stress, especially tensile stress, is generally harmful to the properties of the deposits. For example, the high tensile stress reduces the toughness and fatigue strength of the deposits [4,12]. Therefore, it is essential to control and reduce the residual stress in the thick nickel deposits.

During the nickel electrodeposition, the addition of suitable organic additives in the electrolytes can significantly alter the grain size and texture of deposits to obtain the desired properties, such as hardness, toughness, and corrosion resistance [1,3,5,6,13,14]. The sulfur-containing additives, such as saccharin, are one of the main types of nickel electrodeposition additives and can significantly refine the grain and reduce the residual stress of the deposits [3,12]. However, the use of these additives leads to the incorporation of sulfur in the deposits, which increases their thermal brittleness [15]. The 2-butyne-1,4-diol (C_4_H_6_O_2_, BD), which neither contains N nor S elements, is a class-II brightener [16,17]. Its addition to the electrolytes can lead to the deposition of semi-bright sulfur-free nickel layers, making it suitable for multilayered nickel plating. Additionally, a small usage of BD can significantly change the surface morphologies and microstructure of the deposits, and its reaction products are some simple alcohols and have little impact on subsequent electrodeposition [5], allowing it to be used in industrial applications. During the electrodeposition, the BD molecules in the baths are adsorbed onto the electrode surface by its triple bond and thus strongly inhibit the electro-crystallization of nickel [16,18]. Meanwhile, the adsorbed BD reacts with atomic hydrogen on the surface, which promotes the consumption of hydrogen ions in the solution near the electrode surface, increasing the local pH value and thus changing the inhibitors adsorbed on the surface [16,18]. Therefore, the BD in the electrolytes affects the grain size, texture, defect density, and mechanical properties of electrodeposited nickel [16,19]. Alimadadi et al. [19] systematically investigated the nickel layers electrodeposited from Watts baths containing BD in various concentrations using complementary characterization methods. Their results indicated that the BD contributed to grain refinement at low concentrations, and the texture of the nickel deposits gradually changed to <111>, and the hardness increased with increasing BD concentration [19]. Sakamoto et al. [16] found that the texture of nickel layers deposited at 3 A/dm^2^ changed successively from <110> to <100> and <111> with increasing BD concentration, and the surface morphologies became spiral-type and lozenge-type, while the actual growth plane was always (111). When the electrolytes were stirred, the addition of BD caused a smooth surface and bright appearance of the deposit [20,21] but also increased the risk of crack formation due to the increase in residual tensile stress [20,22].

To reduce the residual tensile stress caused by BD, it is necessary to reduce the use of BD or to add a stress reliever to the electrolytes [3,12]. However, most stress relievers are sulfur-containing compounds, such as saccharin. Chloride ions, which primarily serve as anode activators and increase conductivity in electrolytes, can alter the residual stress of the deposits [4,23]. Additionally, the strong interaction between chloride ions and the metal’s surface affects the adsorption and reaction of the chemical species in the electrolytes [24,25]. For example, chloride ions can replace the absorbed BD on the cathode surface and alter its hydrogenation products, which may affect its efficiency [26]. Therefore, changing the concentration of chloride ions in the electrolytes may help to adjust the effect of BD on the residual stress of the deposits. The presence of chloride ions in the electrolytes has been observed to significantly affect the adsorption mode and action of saccharin [27]. Kolonits et al. [3] observed that the synergistic effects of chloride ions and saccharin facilitated a bimodal grain-size distribution in electrodeposited nickel. However, only a few studies have considered the effect of chloride ions on the efficiency of BD action and their synergistic effects on the microstructure and residual stress of electrodeposited nickel.

The primary aim of this study is to assess the effect of chloride ions on the action of BD, as well as their synergistic effects on electrodeposited nickel. For this, the evolution of the microstructure of nickel deposits with varying concentrations of BD and chloride ions was investigated by using complementary characterization methods. The residual stress and the Vickers hardness (HV) test were conducted to characterize the mechanical properties of the deposits. Thereafter, the growth process of the nickel electrodeposits was analyzed, and the mechanisms of BD and chloride ions affecting the residual stress and hardness were explained via analysis of surface morphologies and microstructure.

## 2. Experimental

### 2.1. Electrodeposition

Nickel deposits were prepared from sulfamate baths via the direct-current electrodeposition technique. Three sulfamate baths (baths A, B, and C) contained the same nickel ion concentration, but varying chloride ion concentrations were used. In these baths, nickel sulfamate (Ni(SO_3_NH_2_)_2_•4H_2_O) was the solute supplying Ni^2+^, and boric acid (H_3_BO_3_) was the pH buffer. Furthermore, nickel chloride (NiCl_2_•6H_2_O) was employed as a chloride ion source and for solving the anode, and sodium dodecyl sulfate (C_12_H_25_SO_4_Na) served as the wetting agent. Table 1 presents the chemical compositions of these baths in detail. Additionally, different BD concentrations were used as additives in the baths. The nickel deposits from the 3 baths, as obtained using a series of combinations of BD and chloride ion concentrations, are presented in Table 2.

During the electrodeposition, sulfur-activated nickel (Ni content > 99.96 wt.% and S content 0.027–0.04 wt.%), which was specifically designed for the process, was used as a soluble anode to supplement the consumption of Ni^2+^ in the baths. Sulfur activation promotes the uniform dissolution of nickel, even in chloride-free baths. Further, a pure copper sheet was selected as the cathode for nickel deposition. The cathode exhibited a surface area of 0.1 dm^2^ (25 mm × 40 mm), which was polished and cleaned before the deposition. The initial pH values of the baths were adjusted to 4.0 by adding sodium hydroxide or sulfamic acid solutions, and the temperatures of the baths were maintained at 323 K at a constant magnetic stirring rate of 800 rpm. All the nickel layers were deposited at a constant current density of 2 A/dm^2^. The electrodeposition was conducted for 2 h to reach a nominal thickness of ~50 μm according to Faraday’s law. The current efficiency was estimated by weighing the cathodes before and after deposition. Deposits with a thickness of 20 μm were also prepared to evaluate the thickness-based change in the residual stress.

### 2.2. Microstructure Characterization

The crystal structure of the nickel deposits was analyzed using X-ray diffraction (XRD, Smartlab 9X) with Cu-Kα radiation (λ = 0.15405 nm). To estimate the preferred orientation of the electrodeposited nickel, the relative texture coefficient RTC_(hkl)_ was calculated from the diffraction intensities of the (hkl) plane using the following equation [5]:RTC_(hkl)_ = (I_hkl_/I^0^_hkl_)/(ƩI_hkl_/I^0^_hkl_ × 100%),(1)
where I_hkl_ and I^0^_hkl_ are the diffraction intensities of the (hkl) planes in the nickel deposits and in nickel powder, respectively. The 4 major peaks, i.e., (111), (200), (220), and (311), were used as the total value of the denominator.

Scanning electron microscopy (SEM, Zeiss Gemini SEM 500) was used to examine the surface morphologies of the nickel deposits at an acceleration voltage of 5 kV and a working distance of 10 mm. Further, the electron back-scattered diffraction (EBSD) images of the nickel deposits along the growth direction (GD) were characterized via SEM at a step size of 0.07 μm. The data was processed using the Channel 5 software (Version: 5.0.9.0). The microstructures of some of the deposits were also investigated via transmission electron microscopy (TEM, FEI Talos F200X) operating at 200 kV. Additionally, an atomic force microscopy (AFM, Bruker Dension Icon) was used to characterize the surface morphology in an area of 10 µm × 10 µm.

### 2.3. Properties Measurements

The residual stress in the nickel deposits was tested via non-destructive XRD techniques using the classical sin^2^ψ method [28]. The peak of the (311) plane was measured by an X-ray diffractometer (X’Pert Pro MPD) with Co-Kα radiation under 35 kV and 30 mA. The PANalytical Stress software (Version: 1.1a) was used to process the XRD data, and the residual-stress values were estimated from the slope of the sin^2^ψ vs. 2θ curves. Moreover, to evaluate the mechanical properties of the various electrodeposits, the Vickers micro-hardness measurement was conducted on the cross-section of the deposits, and a peak load of HV0.05 and a hold time of 10 s were employed for the measurements. The hardness values comprised the average of 6 test values.

## 3. Results

### 3.1. Surface Morphology

The surface morphologies of the nickel layers that were deposited from the sulfamate baths containing varying BD and chloride ion concentrations were characterized by SEM, as shown in Figure 1. Obviously, the morphologies of the nickel deposits (A-0, B-0 and C-0) were similar when no organic additive was utilized in the baths, and chloride ions hardly affected them. Figure 1a,e,i show that the surfaces of these deposits were rather uneven, with pyramidal morphology, which are the typical surface characteristics of low current electrodeposited nickel using baths without additives [1,29]. The pyramidal morphology was formed by means of screw dislocations [30,31]. When a screw dislocation intercepts the surface, a step edge generally emerges at the center of the screw dislocation, which is the most active site of nickel atom deposition. During electrodeposition, the reduced nickel atoms diffuse into the steps and settle at kinks, resulting in crystal growth along the screw dislocation and winding up into pyramids.

After adding a small amount of BD (0.3 mmol/L) to the baths, the surface of the nickel deposits flattened (Figure 1b,f,j). The change reflected the inhibitory effect of BD on nickel electrodeposition. Here, some coarse, unshaped crystals appeared on the deposit surface, although they were poorly defined because of their heavily ridged and irregular faces, and a few small, pyramidal grains were distributed between them. The irregular faces were formed by the adsorption and inhibition of the BD molecules on the surface. Notably, the synergistic effects of BD and chloride ions on the morphology became noticeable from this point. The surface characteristics of Deposit A-0.3 varied and included tiny ridges, sharp edges, and small bumps, indicating its fragmented morphology. Conversely, the surface of Deposits B-0.3 and C-0.3 were mainly characterized by small bumps with ridged faces, which appeared organized.

By increasing the BD concentration to 1 mmol/L, the surface morphologies of the deposits were further altered, and the synergistic effects of BD and chloride ions became evident. Figure 1c shows that the surface of Deposit A-1, which was deposited from Bath A without any chloride ion content, was uneven with several noticeable bumps; these bumps were characterized by microridged faces and beveled sides. The edges of the bumps were sharp and occasionally merged in ridges. However, regarding the nickel layers that were deposited from the chloride ion-containing baths, noticeable grooves appeared on the surface, and the crystals outlined by the grooves were mainly characterized by blunted faces and round edges, as well as partly by ridged faces (Figure 1g,k). These indicate that the chloride ions in the baths affected the inhibition of BD.

By further increasing the BD concentration to 3 mmol/L, the surface morphologies of the nickel deposits from the various baths became consistent again, and their surfaces were covered with tiny bumps, as shown in Figure 1d,h,l. The nickel deposits exhibited more regular and uniform surface structures. The action mode of BD, as well as the growth process of the crystals, changed completely under this condition.

### 3.2. Crystallographic Texture

Figure 2 shows the XRD spectra and RTC_(hkl)_ of the nickel layers that were deposited from different baths. It is easy to find that the main diffraction peaks of the electrodeposits clearly matched with XRD peaks of the reference nickel phase (JCPDS: 00-004-0850). However, the peak exhibiting the highest intensity varied with the BD concentration. In the absence of the additive, the (220) peak exhibited the highest intensity, and its RTC_(220)_ value was >84%, indicating the presence of a strong <110> texture in the nickel deposits. The formation of the <110> texture in the nickel layers was attributed to the inhibition of the surface-adsorbed hydrogen atoms (H_ads_) [32]. Additionally, the RTC_(220)_ value decreased slightly with increasing chloride ion concentration in the baths, indicating a decrease in the amount of H_ads_ on the surface. After adding 0.3 mmol/L BD to the baths, only the (200) peak appeared distinctly and other peaks disappeared almost completely. RTC_(200)_ increased to ~100%, and the deposits exhibited a strong <100> texture. Further, increasing the BD concentration to 1 mmol/L slightly altered the RTC_(200)_ value, and the <100> texture in the deposits was unaffected. The <100> texture corresponds to the free lateral crystal growth mode and is generally associated with coarse grains and low residual stress [1,33]. The addition of 3 mmol/L BD to the baths significantly decreased the (200) peak intensity, and the (111) peak became the main peak; thus, the preferred orientation of the nickel deposits changed from <100> to <111>. The <111> texture was attributed to the inhibited lateral growth, and the inhibition of Ni(OH)_2_ precipitation on the deposit surface resulted in the formation of this texture [32]. Furthermore, the proportions of the <111> texture components in the deposits decreased, and the proportion of the <100> texture component increased with increasing chloride ion concentration in the baths. This indicated a decrease in the Ni(OH)_2_ precipitation on the surface. Put simply, the preferred orientation changed successively from <110> to <100> and <111> with the increase in the BD concentration, and the chloride ions hardly impacted it.

### 3.3. Microstructure

EBSD analysis was performed to study the microstructures of the electrodeposits; the inverse pole figure (IPF) maps of the cross-section of the nickel layers are shown in Figure 3. Figure 3a,d,g reveal that the acicular grains, exhibiting <110> preferred orientation and elongating along GD, reflected the main structural feature of the nickel layers (A-0, B-0, and C-0) that were deposited from the baths without additives. And the acicular grains grew incompletely and exhibited a small aspect ratio. After adding 0.3 mmol/L BD to the baths, the shape of the grains in the deposits changed significantly. Figure 3b,e,h show that several coarse columnar grains with high aspect ratios were presented in the fine-grained matrix. Moreover, the columnar grains and most of the fine grains could be well identified as mainly <100> oriented along GD by EBSD, although a few of the fine grains were <110> or <111> oriented. A twinning relationship existed between the <111> orientated fine grains and the columnar grains. When the baths contained 1 mmol/L BD, the proportion of the columnar grains in the deposits (A-1, B-1, and C-1), as well as the average grain size, increased, and some of the columnar grains ran through the deposit thickness, as shown in Figure 3c,f,i. It is noteworthy that the grains in Depsits B-1 and C-1 grew more fully along BD and in the lateral direction, and the fine grains between columnar grains almost disappeared. While Deposit A-1, prepared from the baths without chloride ions, retained many fine grains between the columnar grains, and the columnar grains were relatively slender. Further, the texture of these grains was dominated by the <100> orientation along GD, and the grains with other orientations almost disappeared.

After adding 3 mmol/L BD in the baths, the grain size in the deposits was reduced significantly, lower than the resolution of EBSD. Therefore, TEM was performed to ensure a clearer observation of the microstructure. Figure 4 shows the TEM images of Deposits A-3 and B-3; the microstructures of C-3 are similar to them. The images revealed that the grains were smaller than 100 nm and that few twins were present in the deposits. The changes in the grain size of the deposits revealed that the addition of high concentrations of BD to the baths significantly changed the growth process of nickel electrodeposition.

### 3.4. Residual Stress and Hardness

The growth of the electrodeposits followed a layer-by-layer process, which generally forms residual stress in the deposits. The residual stress of the deposits was determined via XRD using the sin^2^ψ method. And the deposits with thicknesses of 20 µm and 50 µm were tested to investigate the variation in the residual stress based on the thickness of the deposits, and the results are shown in Figure 5. As can be seen, the nickel deposits, which were obtained from the chloride ion-containing baths, exhibited increased residual tensile stress values, and the values increased with the increasing chloride ion concentration. Meanwhile, the stress values also increased with the BD concentration. And the addition of BD further enhanced the effect of chloride ions on the residual stress. Furthermore, the residual stress values of the nickel deposits decreased with increasing thickness of the deposit, and the stresses of Deposits A-0, A-0.3, and A-1 were even converted into compressive stress after their thickness increased to 50 µm. However, regarding the nickel layers that were deposited from the chloride-free baths, the residual stress values reduced abnormally following the increase of the BD concentration from 0.3 to 1 mmol/L. Finally, when the concentration of BD exceeded 3 mmol/L, the nickel deposits were characterized by extremely high tensile stress. The nickel layers, deposited from the baths with a BD concentration below 3 mmol/L, have low residual stress and relatively rough surface and are suitable as intermediate layers for multilayered nickel electrodeposition.

Hardness is a crucial property of electrodeposited nickel [13]. Figure 6 shows the variation in the hardness of the deposits with BD concentration. Clearly, the addition of BD to the baths significantly affected the hardness of the deposits, while chloride ions had little effect. The hardness of the deposits decreased slightly as their texture changed from <110> to <100>. Thereafter, the hardness increased significantly when the texture changed to <111>. The evolution of the texture was related to the change in the grain size, thus altering the hardness [1]. Notably, the hardness values of the Bath A deposited nickel layers did not change significantly as the texture changed from <110> to <100>.

## 4. Discussion

### 4.1. Influence of the Bath Composition on the Surface Morphology and Microstructure

During nickel electrodeposition, the chemical species in the baths can be adsorbed onto the cathode surface owing to the high surface energy of nickel, and the adsorbed species may significantly affect the nickel deposition process, resulting in changes in the morphology, microstructure, and properties of the deposits [32]. Regarding the nickel layers (A-0, B-0, and C-0) that were electrodeposited from the sulfamate baths without additives, their surfaces exhibited a pyramid morphology, and their microstructures were mainly composed of acicular grains with <110> preferred orientation. At a low current density (2 A/dm^2^), the reduction of Ni^2+^ was accompanied by that of H^+^; thus, the cathode surfaces were covered by a large amount of H_ads_, which promoted the formation of the <110> texture [24,32]. The hydrogen evolution on the surface accounted for the incomplete growth of the acicular grains along GD as well as its thinning. Further, the low current density resulted in the low supersaturation of the nickel adatoms on the electrode surface. Thus, the grains grew with the help of screw dislocation, inducing the appearance of pyramidal morphology. The growth type of these deposits corresponded to the field-oriented isolated crystals type (FI) [34]. In addition, when the baths did not contain chloride ions, the electroactive particle of nickel electrodeposition was NiOH^+^, which exhibited a low concentration in the acidic baths (pH = 4) [35,36]. At this point, the electrode process began with H^+^ reduction, which facilitated the alkalization of the solution near the electrode surface and the formation of NiOH^+^ [35,37]. The hydrogen reduction preceding the metal deposition promoted further H_ads_ adsorbed on the surface, imparting Deposit A-0 with a slightly strengthened <110> texture.

When BD was added to the sulfamate baths, it acted as a strong inhibitor for nickel deposition; its reaction kinetics were controlled by the mass-transport process [19,38,39]. During the electrodeposition, the BD molecules in the baths were adsorbed onto the active sites on the electrode surface, particularly the center of screw dislocation, where they inhibited the deposition of nickel atoms, resulting in the disappearance of the coarse pyramidal morphology. Moreover, these adsorbed BD molecules were continuously consumed through hydrogenation, which was faster than its diffusion from the solution onto the electrode surface [39]. Figure 7 shows that the current efficiency of nickel electrodeposition decreased linearly with the increasing BD concentration, indicating that BD consumption was related to its concentration and that its reaction rate was controlled by diffusion. Meanwhile, the chloride ions barely affected the current efficiency, and the consumption of BD was almost unaffected by chloride ions.

When the BD concentration in the baths was below 3 mmol/L, the deposits developed a <100> texture, the surface was flattened, and the coarse columnar grains appeared and increased in number with increasing BD concentration. The transformation of the texture from <110> to <100> can be explained by the hydrogenation of BD molecules, which rapidly consumed H_ads_ on the surface and caused the free-growth mode to operate. However, the BD molecules adsorbed on the cathode surface and inhibited the deposition of nickel atoms on the sites they occupied. The inhibition strength of BD was related to its coverage on the surface, and they were first adsorbed on (100) planes on the apex of the bumps via diffusion [16]. The higher the concentration of BD in the baths, the more BD adsorbed on the surface, and the stronger the inhibition of out-growth, so the flatter the surface. Meanwhile, the inhibition of BD on the planes that parallel to the surface also promoted grain coarsening. Due to the low concentration of BD or its desorption, there were always some unblocked deposition sites on the surface, which were also readily blocked by BD; thus, many micropeaks remained on the deposit surface, as shown in the AFM images of the deposits (Figure 8). The repeated appearance of the deposition sites on a grain promoted its growth into a columnar grain. The growth type of the deposits corresponded to the field-oriented texture crystals type (FT) [34]. The formation of fine grains between the columnar grains may have been due to the incorporation of residual H_ads_ or hydrogen gas at the grain boundaries. For nickel layers (Deposits A-0.3 and A-1) electrodeposited from the baths without chloride ions, the hydrogen reduction, which occurred prior to the nickel deposition [40,41], may accelerate the hydrogenation of BD. Therefore, the inhibitory effect of BD was weakened, additional available deposition sites appeared on the surface, and increasingly scattered formation and growth of micropeaks were observed in Deposits A-0.3 (Figure 8a). As shown in Figure 9a,c, Deposit A-0.3 exhibited lower roughness and height deviations from the mean line than B-0.3. At higher BD concentrations, the rapid hydrogenation of large amounts of BD promoted the formation of NiOH^+^ near the apex of the peaks and benefitted the out-growth of the micropeaks. Hence, in Figure 8b, Deposit A-1 exhibited evident bumps on its surface. Concurrently, the surface height and the roughness of A-1 were also higher, as shown in Figure 9. The out-growth process of A-1 produced relatively fine columnar grains.

After adding 3 mmol/L BD to the baths, the electrodeposited nickel layers were composed of nanoscale grains and formed a <111> texture, and their surface exhibited a large number of tiny bumps. During the electrodeposition, the hydrogenation of excessive BD molecules consumed extra hydrogen, which caused a rapid increase in the pH near the electrode surface, favoring Ni(OH)_2_ precipitation. Ni(OH)_2_ is considered a very efficient inhibitor of nickel growth, as well as a stabilizer of the <111> texture [19,32]. Under the strong inhibition of Ni(OH)_2_, the growth of the grain stopped frequently, and the deposits grew via repeated nucleation. Therefore, the grains in the deposits were fine, and the surface was flat. The growth type of these deposits corresponded to the unoriented dispersion type (UD) [34]. The higher <111> texture components in Deposit A-3 may be due to the higher instantaneous surface pH caused by the prioritized hydrogen reduction.

### 4.2. Influence of the Bath Composition on Residual Stress and Hardness

The residual stress in the electrodeposited nickel was affected by numerous factors [42,43]. The results indicate that the residual stress in the electrodeposited nickel was enhanced by BD and chloride ions. Tsuru et al. [36] found that the addition of chloride ions to the baths resulted in the formation of NiOH^+^Cl^−^ ion pairs, which were readily included in the deposits due to they were electrically neutral. The subsequent desorption of OH^−^ and Cl^−^ may lead to the shrinkage of the deposits and the formation of residual tensile stress. The higher the chloride ion concentration in the baths, the more the ion pairs would be formed and incorporated in the deposits, and the greater the residual tensile stress of the deposits would be. The hydrogenation of BD promoted the alkalization of the solution near the electrode surface, which may favor the formation of the ion pairs; thus, BD in the baths enhanced the effect of chloride ions on the residual stress.

Further, adding BD to the baths can generally induce an increase in residual tensile stress in the deposits. When the BD concentration was below 3 mmol/L, a strong <100> texture was formed in the deposits. Additionally, the adsorption and inhibition of BD first occurred on the (100) planes [16], which were parallel to the electrode surface. Therefore, both the diffusion of nickel adatoms and the applied current density on these surfaces decreased. Nickel atoms were deposited more on the free (111) surfaces, giving them a higher growth rate and promoting the realization of lateral-type growth. The decreases of the <111> texture component with increasing BD concentration demonstrated in Figure 2 and Figure 3 also prove that the (111) surfaces of the <111> oriented twin grains grew rapidly and disappeared, while the (100) surfaces adsorbed BD and survived with a slow growth rate, making it difficult for the twin grains to grow. Lateral-type growth promoted grain coarsening, which caused the grains to deviate from their equilibrium shape. Therefore, tensile stress in the deposits increased as a consequence of the grain shape and the coalescence of the crystallites following the thermodynamic theory of Pangarov and Pangarova [44,45]. When the concentration of BD was 3 mmol/L, the precipitated Ni(OH)_2_ on the surface was partly incorporated into the deposits, and the deposit shrinkage, which was caused by the desorption of OH^-^, resulted in the formation of high tensile stress [11]. The decrease in the residual stress with the increase in thickness is due to the stress relaxation and BD concentration reduction caused by the substrate bending and BD consumption, respectively. The higher thermal expansion coefficients of copper than that of nickel resulted in the appearance of compressive stress in Deposits A-0, A-0.3, and A-1 after the thickness increased to 50 µm [10]. Moreover, for Deposit A-1, the lateral type growth of Deposit A-1 was weak, and the crystal shape deviated slightly from the equilibrium shape, resulting in the formation of small residual stress. Meanwhile, the high surface roughness and BD concentration promoted the incorporation of impurities [46]; thus, A-1 gradually developed evident compressive stress with increasing thickness.

It is well known that the hardness of materials is closely related to their grain size and that a Hall–Petch relationship exists between them [3,47]. The grain sizes of the deposits were evaluated from the broadening of the XRD peaks via the Scherrer equation [48]. The values that are obtained by this method are generally low because they represent the crystal size of the coherent diffraction domains [48]. Figure 10 shows that the hardness was plotted as a line function of the inverse square root of the crystal size, *d*. The deviation of partial points may be due to the influences of defect density, texture, and grain boundary structure [49]. Although the crystal size may be underestimated, the small size also reflects that the deposits contained numerous defects and that the deposit growth proceeded through the repeated formation and merging of micropeaks. In the absence of any additive in the baths, the hydrogen evolution on the surface refined the crystals. After adding 0.3 mmol/L BD to the electrolytes, the hydrogen evolution was replaced by BD hydrogenation, the crystal refinement was weakened, and the hardness was reduced. Regarding Deposit A-0.3, the activation of additional active sites reduced and increased the crystal size and hardness, respectively. When the BD concentration in the baths increased to 1 or 3 mmol/L, the increased BD or Ni(OH)_2_ adsorption resulted in the increased interference of the deposition of the active sites, forming additional defects and further reducing the crystal size, and continuously increasing the hardness.

## 5. Conclusions

Herein, the microstructure and residual stress of electrodeposited nickel layers, which were prepared from the sulfamate baths containing varying BD and chloride ion concentrations, were tested. The individual and synergistic effects of BD and chloride ions were explained via analysis of the results. The key conclusions are as follows:The addition of BD to the sulfamate baths resulted in the replacement of the <110> oriented acicular grains by the <100> oriented coarse columnar grains or <111> oriented nanograins, as well as the increases in the residual stress;For BD-containing baths, BD molecules were adsorbed on the cathode surface and inhibited nickel deposition. And the grains in the deposits grew through the continuous formation and merging of the micropeaks on the surface. These processes determined the surface morphologies and microstructure, therefore affecting the residual stress and hardness of the deposits;Adding chloride ions into the baths could improve the inhibition of BD, which slightly promoted surface flattening and the <100> oriented grain coarsening, but evidently increased the residual stress in the nickel layers. The BD enhanced the effect of chloride ions on the residual stress. Therefore, reducing the concentration of chloride ions in the baths is a feasible way to obtain a low residual stress nickel deposit when using BD-containing baths for electrodeposition.

## Figures and Tables

**Figure 1 materials-16-03598-f001:**
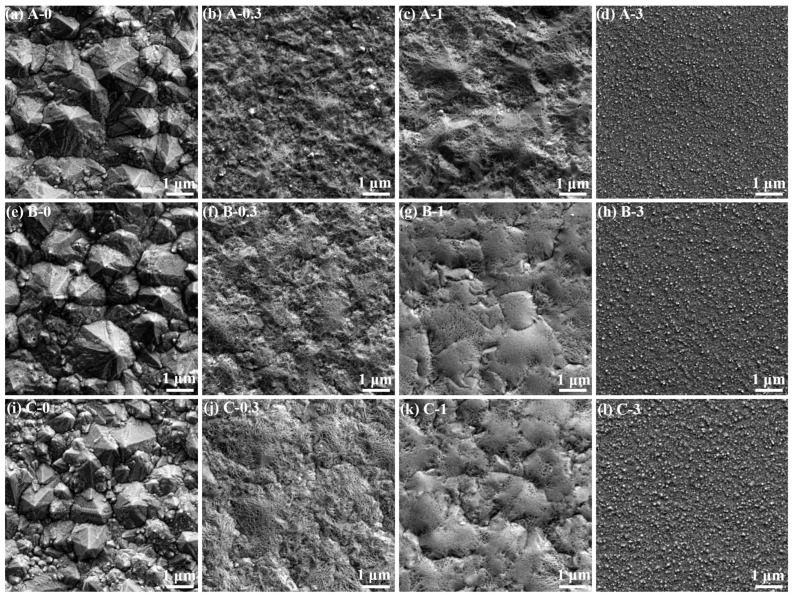
SEM images of the surface morphologies of the nickel layers that were electrodeposited from the sulfamate baths containing varying BD and chloride ion concentrations.

**Figure 2 materials-16-03598-f002:**
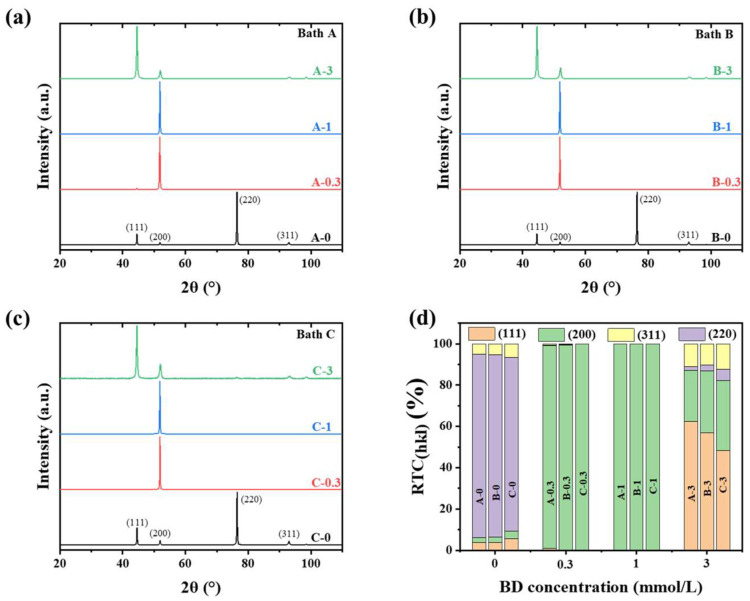
(**a**–**c**) XRD patterns of the nickel layers that were electrodeposited from Bath A, B, and C, respectively. (**d**) RTC values of the various (hkl) planes of each nickel layer.

**Figure 3 materials-16-03598-f003:**
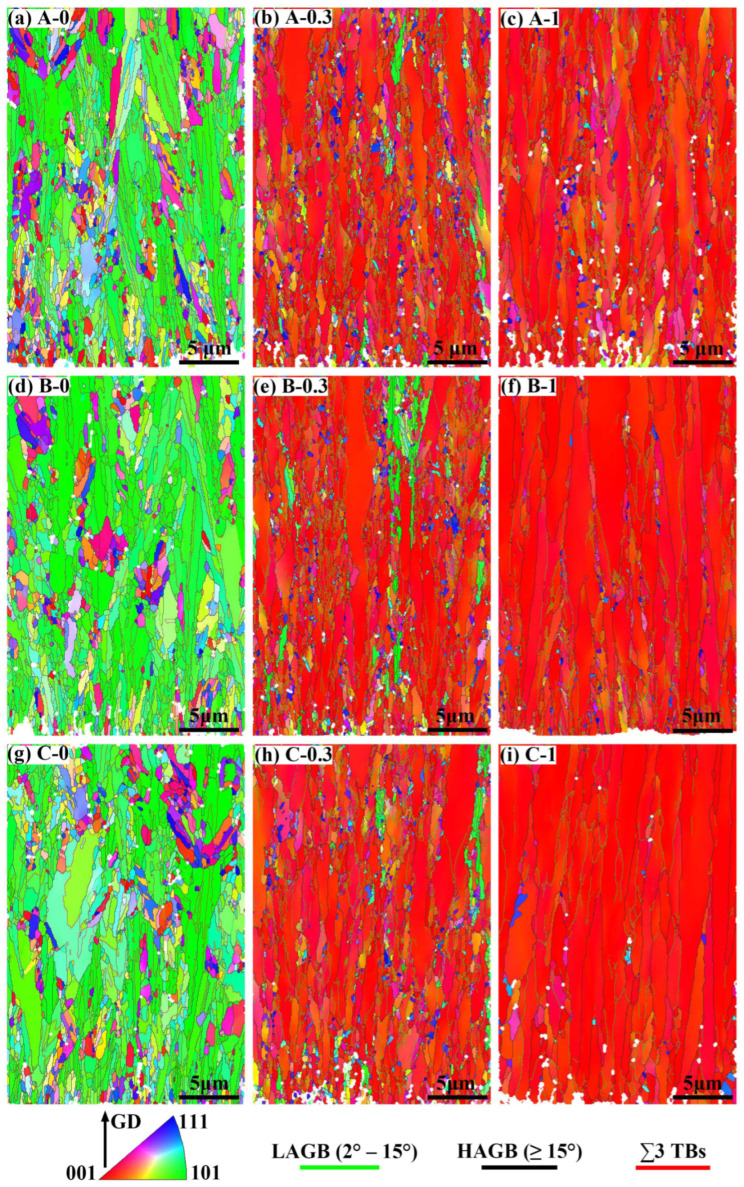
IPF maps of the cross-sections of the various electrodeposited nickel. The grains are colored by GD using the IPF convention in the inset. The light green, black and red colors represent the low-angle grain boundaries (2°–15°), high-angle grain boundaries (≥15°), and ∑3 twin boundaries, respectively.

**Figure 4 materials-16-03598-f004:**
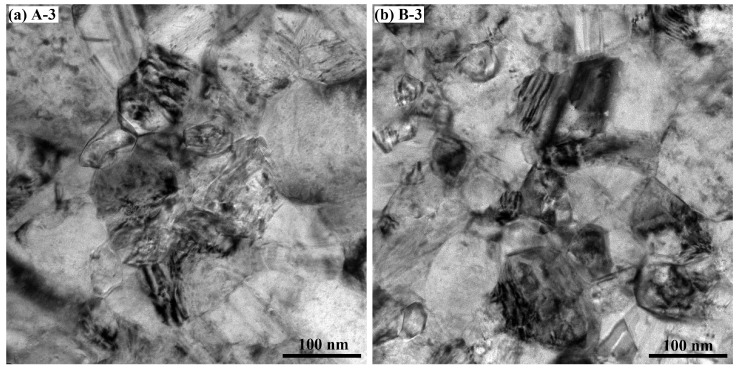
TEM images of nickel deposits (**a**) A-3 and (**b**) B-3.

**Figure 5 materials-16-03598-f005:**
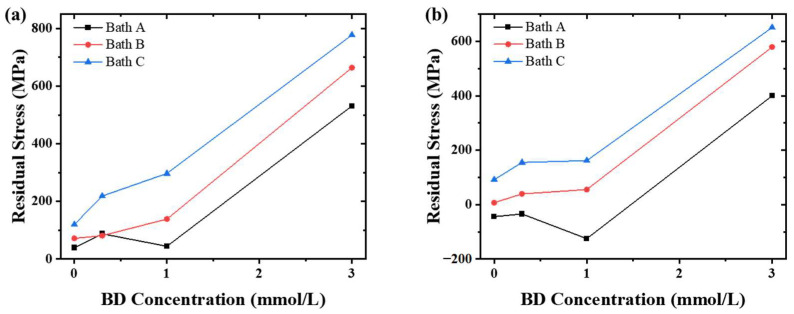
Variation of the residual stress of the nickel deposits with the BD concentrations of the baths: (**a**) 20 and (**b**) 50 μm nickel deposits.

**Figure 6 materials-16-03598-f006:**
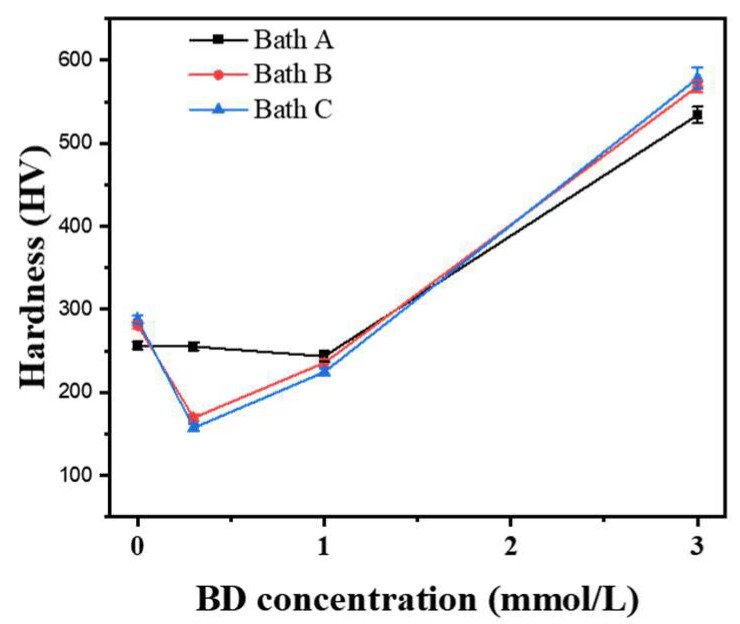
Variation of the HV of the nickel deposits with the BD concentrations of the baths.

**Figure 7 materials-16-03598-f007:**
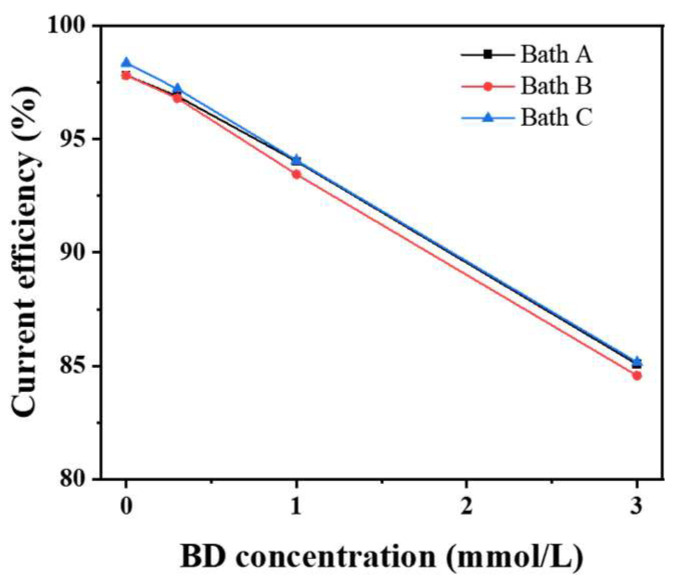
Variation of the current efficiency of the nickel deposits with the BD concentrations of the baths.

**Figure 8 materials-16-03598-f008:**
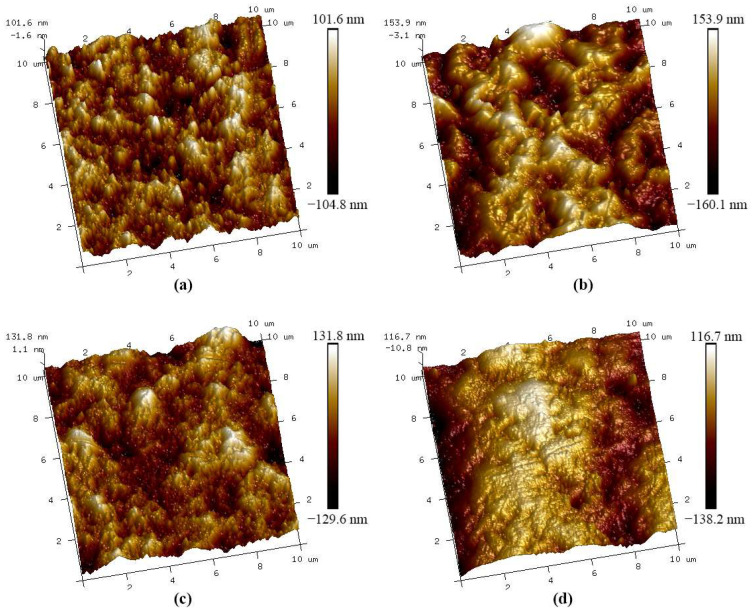
AFM images of the nickel deposits: (**a**) A-0.3, (**b**) A-1, (**c**) B-0.3, and (**d**) B-1.

**Figure 9 materials-16-03598-f009:**
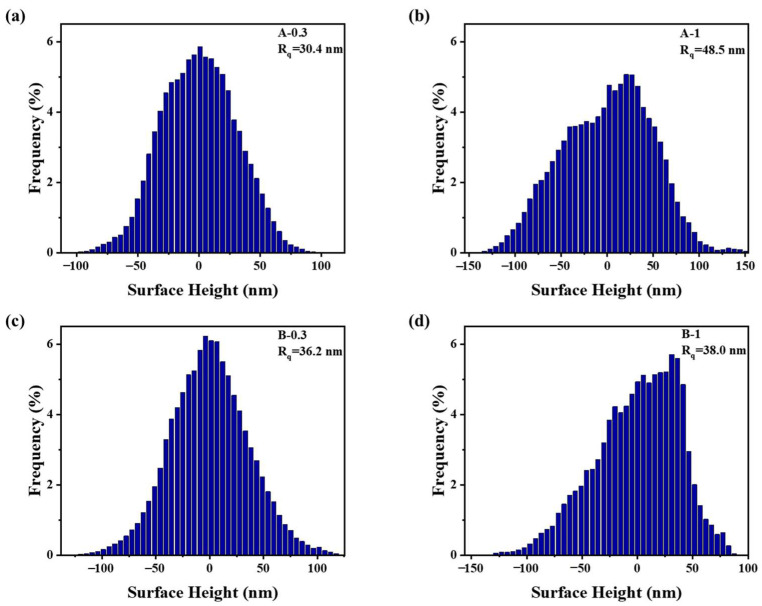
AFM roughness histograms of (**a**) A-0.3, (**b**) A-1, (**c**) B-0.3 and (**d**) C-1. R_q_ represents the root mean square roughness.

**Figure 10 materials-16-03598-f010:**
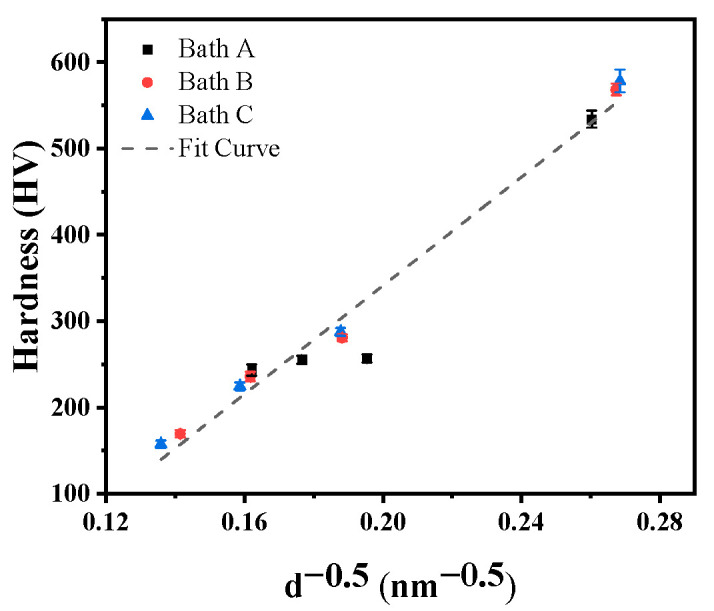
Hall-Petch plot for the HV vs. the inverse square root of the crystal size (*d*).

**Table 1 materials-16-03598-t001:** Composition of the employed sulfamate baths (unit: g/L).

Composition	Bath A	Bath B	Bath C
Ni(SO_3_NH_2_)_2_•4H_2_O	350	336	309
NiCl_2_•6H_2_O	0	10	30
H_3_BO_3_	40	40	40
C_12_H_25_SO_4_Na	0.1	0.1	0.1

**Table 2 materials-16-03598-t002:** Electrodeposited nickel layers from three sulfamate baths (Baths A, B, and C) with different BD concentrations (0, 0.3, 1, and 3 mmol/L).

Bath	BD Concentration (mmol/L)			
	0	0.3	1	3
A	A-0	A-0.3	A-1	A-3
B	B-0	B-0.3	B-1	B-3
C	C-0	C-0.3	C-1	C-3

## Data Availability

Not applicable.

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
