# Peer review of "Synergistic Effects of 2-Butyne-1,4-Diol and Chloride Ions on the Microstructure and Residual Stress of Electrodeposited Nickel"

_materials, 2023, doi:10.3390/ma16093598_

Round 1

Reviewer 1 Report

The paper presented the effects of 2-butyne-1,4-diol and chloride ions on the microstructure and residual stress of electrodeposited nickel. The paper is of good quality, however, there are a few comments:

  1. Most of the references are old. Please update your references.
  2. In the experimental part, how can you obtain or calculate the thickness?
  3. Some references that show how the additives could alter the electrodeposition of Ni (Int. J. Electrochem. Sci., 10 (2015) 4946 – 4971), Surface and Coatings Technology, Volume 347, 2018, Pages 113-122, https://doi.org/10.1016/j.surfcoat.2018.04.079.), Int. J. Electrochem. Sci., 17 (2022) Article Number: 22044, doi: 10.20964/2022.04.12.

No comments

Author Response

Dear reviewer,

Thank you very much for taking your time to review this manuscript. We really appreciate all your comments and suggestions. Now we have revised the manuscript according to the comments. Most of the revisions in the manuscript are marked in red. Explanations regarding to these revisions are as follows.

1. Most of the references are old. Please update your references.

Answer: Thanks for the comments. We updated some of the old references and added a few recent references. The updated references include: [8], [9], [13], [14], [21], [22], [31], [37], [38].

2. In the experimental part, how can you obtain or calculate the thickness?

Answer: Thanks for the comments and we added a description on how to calculate the thickness in section 2.1.

As follow:

All the nickel layers were deposited at a constant current density of 2 A/dm2. The electrodeposition was conducted for 2 h to reach a nominal thickness of ~50 μm according to Faraday’s law.

3. Some references that show how the additives could alter the electrodeposition of Ni. (Int. J. Electrochem. Sci., 10 (2015) 4946 – 4971), (Surface and Coatings Technology, Volume 347, 2018, Pages 113-122, https://doi.org/10.1016/j.surfcoat.2018.04.079.), (Int. J. Electrochem. Sci., 17 (2022) Article Number: 22044, doi: 10.20964/2022.04.12.).

Answer: Thanks for the comments and we read these references. The contents of them are helpful for us to understand the role of additives in nickel electrodeposition, and citing them in our manuscript can better demonstrate the ways in which additives act during the electrodeposition process.

Reviewer 2 Report

In the current Manuscript the individual and synergistic effects of 2-butyne-1,4-diol (BD) and chloride ions on the microstructure and residual stress of electrodeposited nickel have been analyzed and discussed. The relevance of the research finding is important for the researchers working in the field of Ni-electroplating.

Authors should must address the following concerns in the revised manuscript-

(i) Please provide the recent references highlighting the effect of BD on thickness, morphology, hardness and other textural appearances.

(ii) How the current Manuscript provide the novel approach with respect to already adopted procedures by other researchers?

(iii) How the Authors addressed the gaps in the literature?

(iv) Please briefly explain how the use of BD would affect positively on the Industrial application with respect to cost and ease of operation.

(v) Please suggest, Whether use of BD will helpful in multilayered Ni Plating o not?

(vi) Whether the Ni plated substrate discussed in the current Manuscript may be used for further next layer electroplating or not?

Author Response

Dear reviewer,

Thank you very much for taking your time to review this manuscript. We really appreciate all your comments and suggestions. Now we have revised the manuscript according to the comments. Most of the revisions in the manuscript are marked in red. Explanations regarding to these revisions are as follows.

1. Please provide the recent references highlighting the effect of BD on thickness, morphology, hardness and other textural appearances.

Answer: Thanks for the comments. In section 1, we added some contents to introduce recent research and reports on the effects of BD on morphology, hardness, texture, and microstructure.

As follow:

Alimadadi et al. systematically investigated the nickel layers electrodeposited from Watts baths containing BD in various concentrations using complementary characterization methods. Their results indicated that BD in the electrolytes contributed to grain refinement at low concentration, and the texture gradually changed to <111> and the hardness increased with increasing BD concentration. Sakamoto et al. found that the texture of nickel layers deposited at 3 A/dm2 changed successively from <110> to <100> and <111> with increasing BD concentration, and the surface morphology transformed into spiral-type and lozenge-type, while the actual growth plane was always (111). Moreover, the addition of BD caused a smooth surface and bright appearance of the deposits, but also increased the risk of crack formation, due to the increase in residual tensile stress.

2. How the current Manuscript provide the novel approach with respect to already adopted procedures by other researchers?

Answer: Thanks for the comments.

The results in the current Manuscript indicate that the chloride ions in the electrolytes containing BD slightly promote surface flattening and grain coarsening, but significantly increase the residual stress. Therefore, altering the concentration of chloride ions in the electrolytes can adjust the residual stress in the deposits. In addition, using sulfur-activated nickel as anode ensures its uniform dissolution during the nickel electrodeposition, even in chloride-free baths.

3. How the Authors addressed the gaps in the literature?

Answer: Thanks for the comments.

The chloride ions in the electrolytes have a strong interaction with the metal surface, which may affect the adsorption and action of additive. However, there are only a few literatures that have considered the effect of chloride ions on the efficiency of additive, and the synergistic effects of BD and chloride ions on the microstructure and residual stress of nickel deposits have not been studied before. For clarifying theses, we prepared a series of nickel layers by electrodeposition from sulfamate baths comprising varying concentration of BD and chloride ions. The microstructure and properties of these deposits was investigated using SEM, XRD, EBSD, TEM, AFM, and Vickers hardness tester. The synergistic effects of them were explained via analysis of surface morphology, texture, microstructure, residual stress, and hardness.

4. Please briefly explain how the use of BD would affect positively on the Industrial application with respect to cost and ease of operation.

Answer: Thanks for the comments and we added the explain in the section 1.

As follow:

The 2-butyne-1,4-diol, which neither contains N nor S elements, is one of the most commonly employed leveling and brighter for sulfur-free nickel electrodeposition. Adding a small amount of BD to the electrolytes can significantly change the surface morphology and microstructure of the deposits, and the reaction products of BD are some simple alcohols, which have little impact on subsequent electrodeposition.

5. Please suggest, whether use of BD will helpful in multilayered Ni Plating or not?

Answer: Thanks for the comments. The use of BD will be helpful in multilayered nickel plating.

The BD additive belongs to class-II brightener, and adding BD alone into the electrolytes will lead to the production of semi-bright nickel plating. Meanwhile, the use of BD can’t cause sulfur to be incorporated into the nickel layers. Hence, BD can be used for semi-bright nickel plating in multilayered nickel plating.

6. Whether the Ni plated substrate discussed in the current Manuscript may be used for further next layer electroplating or not?

Answer: Thanks for the comments. The nickel plated substrate could be used for further next electroplating.

The nickel layers electrodeposited from the electrolytes containing low concentrations of BD is semi-bright, and has low residual stress and <100> texture, which is associated with high plasticity. And BD is usually employed in combination with sulfur-containing additive to product bright nickel plating Therefore, the nickel plated substrate can be directly used for further next layer electroplating after without cleaning.

Reviewer 3 Report

materials-2366117

Title: Synergistic effects of 2-butyne-1,4-diol and chloride ions on the 2 microstructure and residual stress of electrodeposited nickel

Ming Sun, Chao Zhang, Ruhan Ya, Hongyu He, Zhipeng Li, Wenhuai Tian

The manuscript reports the study of the effect of chlorine ions on the structure and residual stresses of electrodeposited nickel layers. The authors of the peer-reviewed manuscript suggested that the presence of chloride ions may enhance the positive and reduce the negative effects of 2-butyne-1,4-diol on the smoothness, hardness, and residual stress of such layers. The idea of the study is justified. The experimental methods used by the authors are appropriate to the problem to be solved.

The obtained results showed that the assumption of the authors turned out to be erroneous. The presence of chloride ions does not significantly affect the smoothness and hardness of the layers, the properties to be improved. In contrast, it causes an increase in the residual stress of the layers, the value of which must be reduced. The negative results of the study do not mean that the study was useless. However, both the abstract and the conclusion of the manuscript must accurately and clearly explain the essence of the matter.

I have critical remarks both on the style of presentation and on the scientific part of the manuscript.

1.     “The chloride ions in the baths delayed the hydrogenation and desorption of BD, thereby improving their inhibitory ability.” The hydrogenation of 2-butyne-1,4-diol was not studied in the manuscript.

3.     “2. For BD containing baths, BD molecules were adsorbed on the active sites and inhibited nickel deposition.“ The adsorption centers were not studied in the manuscript.

4.     “3. Chloride ions delayed the hydrogenation and desorption of BD, thereby improving its inhibitionand promoting the <100> oriented grain coarsening. Additionally, chloride ions facilitated the inclusion of NiOH+Cl- group into the deposits, which was also enhanced by the hydrogenation of BD. The subsequent desorption of OH- and Cl- increased the residual tensile stress in the deposits.” The kinetic of the hydrogenation and desorption of BD was not studied in the manuscript. Neither the inclusion of NiOH+Cl- nor the desorption of OH- and Cl- were studied in the manuscript.

“Meanwhile, the surface-adsorbed BD molecules are hydrogenation, which improves the pH value of the solution near the electrode surface and alters the inhibitors adsorbed on the surface [13, 15].” Correct both the grammar and the meaning of the sentence. What pH is preferable in this case?

Author Response

Dear reviewer,

Thank you very much for taking your time to review this manuscript. We really appreciate all your comments and suggestions. Now we have revised the manuscript according to the comments. The adstract and the conclusion in the manuscript have been re-written to ensure a clear and accurate introduction of our study. And we also revised some of the sentences in the discussion Most of the revisions in the manuscript are marked in red. Explanations regarding to these revisions are as follows.

1. “The chloride ions in the baths delayed the hydrogenation and desorption of BD, thereby improving their inhibitory ability.” The hydrogenation of 2-butyne-1,4-diol was not studied in the manuscript.

Answer: Thanks for the comments and we have re-written this part of the abstract.

The reaction kinetics of BD are controlled by the mass-transport process. And the current efficiency also indicates that the consumption of BD during the electrodeposition is only affected by its concentration in the bath. The enhanced inhibitory effect of BD in chloride containing baths may be due to its longer adsorption time.

2. "For BD containing baths, BD molecules were adsorbed on the active sites and inhibited nickel deposition. “The adsorption centers were not studied in the manuscript.

Answer: Thanks for the comments and we have re-written this part of the conclusion.

The inhibition of BD is manifested in its adsorption and blocking of the active sites of nickel deposition.

3. "Chloride ions delayed the hydrogenation and desorption of BD, thereby improving its inhibition and promoting the <100> oriented grain coarsening. Additionally, chloride ions facilitated the inclusion of NiOH+Cl- group into the deposits, which was also enhanced by the hydrogenation of BD. The subsequent desorption of OH- and Cl- increased the residual tensile stress in the deposits.” The kinetic of the hydrogenation and desorption of BD was not studied in the manuscript. Neither the inclusion of NiOH+Cl- nor the desorption of OH- and Cl- were studied in the manuscript.

Answer: Thanks for the comments and we have re-written this part of the conclusion.

The enhanced inhibitory effect of BD in chloride containing baths may be due to its longer adsorption time.

Tsuru et al. studied the effects of chloride, bromide and iodide ions on internal stress in films deposited from a nickel sulfamate bath. Their results indicate that the chloride ions in the bath facilitate the incorporation of NiOH+Cl- group into the deposits, which is the reason for the increase in the residual tensile stress. (DOI: 10.1023/A:1003970925918)

“Meanwhile, the surface-adsorbed BD molecules are hydrogenation, which improves the pH value of the solution near the electrode surface and alters the inhibitors adsorbed on the surface [13, 15].” Correct both the grammar and the meaning of the sentence. What pH is preferable in this case?

Answer: Thanks for the comments and we have re-written this sentence. During nickel electrodeposition, the electrode-surface pH value is different from the bulk pH value. The electrode surface pH determines the types and quantity of chemical species near the electrode surface, such as Ni(OH)2, H2 or Hads, thus affecting the inhibitors that adsorbed on the electrode surface.

The revised sentence: 

Meanwhile, the surface-adsorbed BD reacts with atomic hydrogen on the surface. The hydrogenation of BD promotes the consumption of hydrogen ions in the solution near the electrode surface, increasing the local pH value, and thus changing the inhibitors adsorbed on the surface.

Round 2

Reviewer 2 Report

Please incorporate all the changes to revised manuscript at appropriate places. 

Please recheck Manuscript throughly for any error in writing and rephrase if require.

Author Response

Dear reviewer,

Thank you very much for taking your time to review this manuscript again. We really appreciate all your comments and suggestions. Now we have revised the manuscript according to your review report (Round 1). We have added all the changes to the manuscript at appropriate places, and most of the revisions in the manuscript are marked in red. And we have also rechecked the manuscript thoroughly for any error in writing and rephrase if necessary. Explanations regarding to these revisions are as follows.

  1. Please provide the recent references highlighting the effect of BD on thickness, morphology, hardness and other textural appearances.

Answer: Thanks for the comments. In section 1 (Introduction, line 58), we have added some contents to introduce recent research and reports on the effects of BD on morphology, hardness, texture, and microstructure.

As follow:

Alimadadi et al. systematically investigated the nickel layers electrodeposited from Watts baths containing BD in various concentrations using complementary characterization methods. Their results indicated that the BD contributed to grain refinement at low concentration, and the texture of the nickel deposits gradually changed to <111> and the hardness increased with increasing BD concentration. Sakamoto et al. found that the texture of nickel layers deposited at 3 A/dm2 changed successively from <110> to <100> and <111> with increasing BD concentration, and the surface morphologies became spiral-type and lozenge-type, while the actual growth plane was always (111). When the electrolytes were stirred, the addition of BD caused a smooth surface and bright appearance of the deposit, but also increased the risk of crack formation, due to the increase in residual tensile stress.

  1. How the current Manuscript provide the novel approach with respect to already adopted procedures by other researchers?

Answer: Thanks for the comments. The results in the current Manuscript indicate that the chloride ions in the electrolytes can slightly promote surface flattening and grain coarsening, and significantly increase the residual stress. Therefore, altering the concentration of chloride ions in the electrolytes can adjust the effect of additives as well as the residual stress of the deposits. And we have also provided the descriptions in the abstract and conclusion.

As follow:

(Abstract, line 20) And the results indicate that the reduction of chloride ion concentration is a feasible way to reduce the residual stress of the nickel deposits when BD is included in the baths.

(Conclusions, line 443) Therefore, reducing the concentration of chloride ions in the baths is a feasible way to obtain a low residual stress nickel deposit when using BD containing baths for electrodeposition.

  1. How the Authors addressed the gaps in the literature?

Answer: Thanks for the comments. We have added some contents to introduce how we addressed the gaps in the literature.

As follow:

(Introduction, line 85) The primary aim of this study is to assess the effect of chloride ions on the action of BD, as well as their synergistic effects on electrodeposited nickel. For this, the evolution of the microstructure of nickel deposits with varying concentration of BD and chloride ions was investigated by using complementary characterization methods. The residual stress and the Vickers hardness (HV) test were conducted to characterize the mechanical properties of the deposits. Thereafter, the growth process of the nickel electrodeposits was analyzed, and the mechanisms of BD and chloride ions affecting the residual stress and hardness were explained via analysis of surface morphologies and microstructure.

(Conclusions, line 441) Adding chloride ions into the baths could improve the inhibition of BD, which slightly promoted surface flattening and the <100> oriented grain coarsening, but evidently increasing the residual stress in the nickel layers. The BD enhanced the effect of chloride ions on the residual stress. Therefore, reducing the concentration of chloride ions in the baths is a feasible way to obtain a low residual stress nickel deposit when using BD containing baths for electrodeposition.

  1. Please briefly explain how the use of BD would affect positively on the Industrial application with respect to cost and ease of operation.

Answer: Thanks for the comments and we added the explain in the section 1 (Introduction, line 48).

As follow:

Additionally, a small usage of BD can significantly change the surface morphologies and microstructure of the deposits, and its reaction products are some simple alcohols and have little impact on subsequent electrodeposition, allowing it to be used in industrial application.

  1. Please suggest, whether use of BD will helpful in multilayered Ni Plating or not?

Answer: Thanks for the comments. The BD additive belongs to class-II brightener, and adding BD alone into the electrolytes will lead to the production of semi-bright nickel plating. Meanwhile, the use of BD can’t cause sulfur to be incorporated into the nickel layers. Hence, BD can be used for semi-bright nickel plating in multilayered nickel plating. And we have added the descriptions in the section 1 (Introduction, line 45).

As follow:

The 2-butyne-1,4-diol (C4H6O2, BD), which neither contains N nor S elements, is a class-II brightener. Its addition in the electrolytes can lead to the deposition of semi-bright sulfur-free nickel layers, making it suitable for multilayered nickel plating.

  1. Whether the Ni plated substrate discussed in the current Manuscript may be used for further next layer electroplating or not?

Answer: Thanks for the comments. The nickel plated substrate could be used for further next electroplating. Due to the nickel layers electrodeposited from the electrolytes containing low concentrations of BD is semi-bright, and has low residual stress and <100> texture, which is associated with high plasticity. Therefore, the nickel plated substrate can be directly used for further next layer electroplating. And we have added the descriptions in the section 3.4 (Line 274).

As follow:

The nickel layers, deposited from the baths with a BD concentration below 3 mmol/L, have low residual stress and relatively rough surface, and is suitable as intermediate layers for multilayered nickel electrodeposition.